# Carbon Dots Boost dsRNA Delivery in Plants and Increase Local and Systemic siRNA Production

**DOI:** 10.3390/ijms23105338

**Published:** 2022-05-10

**Authors:** Josemaría Delgado-Martín, Alejo Delgado-Olidén, Leonardo Velasco

**Affiliations:** 1Instituto Andaluz de Investigación y Formación Agraria (IFAPA), Churriana, 290140 Malaga, Spain; josemariadm93@gmail.com (J.D.-M.); delgado.oliden@gmail.com (A.D.-O.); 2Universidad de Málaga, 29010 Malaga, Spain

**Keywords:** carbon dots, hydrothermal synthesis, plants, dsRNA delivery, RNA silencing, systemic RNAi, siRNA

## Abstract

In this work, we obtained carbon dots from glucose or saccharose as the nucleation source and passivated them with branched polyethylenimines for developing dsRNA nanocomposites. The CDs were fully characterized using hydrodynamic analyses, transmission electron microscopy, X-ray photoelectron spectroscopy and Fourier transform infrared spectroscopy. The ζ potential determined that the CDs had positive charges, good electrophoretic mobility and conductivity, and were suitable for obtaining dsRNA nanocomposites. DsRNA naked or coated with the CDs were delivered to leaves of cucumber plants by spraying. Quantitation of the dsRNA that entered the leaves showed that when coated with the CDs, 50-fold more dsRNA was detected than when naked dsRNA. Moreover, specific siRNAs derived from the sprayed dsRNAs were 13 times more abundant when the dsRNA was coated with the CDs. Systemic dsRNAs were determined in distal leaves and showed a dramatic increase in concentration when delivered as a nanocomposite. Similarly, systemic siRNAs were significantly more abundant in distal leaves when spraying with the CD-dsRNA nanocomposite. Furthermore, FITC-labeled dsRNA was shown to accumulate in the apoplast and increase its entry into the plant when coated with CDs. These results indicate that CDs obtained by hydrothermal synthesis are suitable for dsRNA foliar delivery in RNAi plant applications.

## 1. Introduction

RNA interference (RNAi) refers to natural defense and regulatory mechanisms of gene expression that were discovered in nematodes in 1998, and since then great progress has been made in its study and applications in plant systems and other biological systems [1,2,3,4]. The presence of exogenous double-stranded RNA (dsRNA) elicits RNAi through the activation of the Dicer proteins and the RNA-induced silencing complex (RISC) that process and use dsRNA as the template for the degradation of complementary RNAs [5,6]. In plants, this biological process is one universal defense mechanism by which plants cope with, e.g., virus infections [7]. The dsRNAs are processed by the RNAi machinery into small RNA molecules, the 21–24-nt short interfering RNAs (siRNAs), that in turn direct the targeting to homologous RNA molecules [8]. In the last years, the topical application of RNA in the form of dsRNAs or siRNAs is emerging as a promising tool in agriculture for the control of pathogens and pests by RNAi, to be potentially included in biological control strategies [9,10,11,12,13]. Once generated, the siRNAs move from plant cells through the plasmodesmata to other 10–15 neighboring cells, in a non-cell autonomous process [8]. In contrast, long RNA molecules (that include mRNAs, tRNAs, and probably dsRNAs), move distantly through the phloem or xylem vessels and from here enter the cells again [3,14,15,16].

For the foliar application of any molecule to the plant, be it a pesticide, a biomolecule or a nutrient, a series of factors such as penetration, stability, and diffusion in the plant must be considered. SiRNAs and dsRNAs have been applied in plants for gene silencing, fungal, virus, and insect control [12,15]. In most cases, these nucleic acids have been delivered naked, in the aqueous phase or buffered [9]. There are examples of their application by spraying at higher or lower pressure [17,18] or mechanically (rubbing) with or without abrasives [19,20]. In the case of plant viruses, several successful cases of spray-induced RNAi control have been described [21], in general, performed under laboratory conditions and more recently in greenhouse conditions [22]. The application of dsRNA or siRNA molecules on plants has also been considered with the aim of being sucked by the harmful insects that feed on them and thus exert a control effect by RNAi [10]. In all these situations, increasing the amount of dsRNA/siRNA entering the plant or improving their internal diffusion will potentially lead to higher efficacy and/or require a smaller amount of them to be effective in RNAi applications [23].

Cell walls are structures of fundamental polysaccharide nature that, in addition to forming the physical structure of the plant cell by surrounding the cell membrane, act as a barrier to the diffusion of molecules, pathogenic organisms, and other environmental agents, including nanoparticles (NPs) [24]. NPs, due to their nanometric scale, possess chemical, surface, and photoelectric properties very different from the same materials at a larger scale that make them suitable for loading and controlled release of active compounds into the plant [25]. Thus, some authors have proposed the use of NPs to improve the delivery conditions of biomolecules to the cellular interior and to facilitate their release in a controlled manner [26,27]. For example, LDH nanoparticles have been used to facilitate or controllably release biomolecules including DNA and dsRNA [17,28,29]. Mesoporous silica nanoparticles have facilitated DNA entry to the point of requiring 1000 times less DNA when bound to these nanoparticles than when released naked to exert gene silencing [28]. Nanoparticles derived of carbon dots (CDs) have facilitated the entry of siRNA into *N. benthamiana* 16c transgenic plants to produce systemic silencing in the plant [18]. On the other hand, the translocation of nucleic acid molecules, once applied to the plant, is as important as or even more important than the entry through the cuticle itself. It has been observed that NPs larger than the usually considered exclusion size of 5–20 nm are able to translocate to the apoplast [30]. However, other reports indicate that the exclusion limit in certain conditions could be higher than that [24]. Undoubtedly, nucleic acid molecules coated with NPs exceed this apparent exclusion limit. In any case, the translocation process of dsRNAs is poorly understood, and even more so the long-distance movements of these nanoparticles when bound to nucleic acids.

CDs are usually 1–10 nm in size, allowing them to pass through the cell wall. Furthermore, they can be synthesized with positive charges, allowing them to electrostatically bind nucleic acids, including dsRNA, have low polydispersity index, and are biocompatible, showing low toxicity [18,31,32]. There are two general methods for CD synthesis, referred to as “top-down” which is characterized by the cleavage of graphitic materials to form CDs, and “bottom-up” which consists of the polymerization using small carbon-containing molecules as precursors [33]. “Bottom-up” methods are, in general, more accessible and include different methods to carry out CD synthesis such as solvo-hydrothermal synthesis, in addition to synthesis by electrochemical methods, microwave-assisted synthesis or laser ablation [34]. In order to obtain positively charged particles, different molecules can be used to passivate the CDs, such as ethanolamine, ethylenediamine, or the widely used polyethylenimines (PEI), either during the process of CD synthesis or after the synthesis by electrostatic union. It has been reported how the passivation of PEI during the synthesis generates the incorporation of nitrogen to the CD backbone and –NH_2_ to the particle surfaces, generating positive charges [34,35]. When exposed to UV light, and depending on their size, CDs emit fluorescence ranging from blue to red [36]. Other key characteristics of CDs are their good water solubility, chemical stability, and resistance to photobleaching [37]. Free CDs are not cytotoxic at high concentrations but when passivated with PEI this threshold is significantly reduced, which is an important feature to consider in biological assays [32]. CDs have found numerous applications in biomedicine and plant biotechnology. They have been used for magnetic resonance imaging (MRI), X-ray and ultrasound bioimaging, targeted drug delivery, biomolecule delivery, biosensors, and theragnostics, among others [32,37,38].

In this work we developed carbon dots to investigate the entry of CD-coated dsRNA into plants, its systemic movement, and that of the derived siRNAs. This study focuses on synthesizing, characterizing, and assessing the feasibility of glucose-derived CDs for dsRNA delivery using cucumber plants as a model. We synthesized positively charged CDs using solvo-hydrothermal synthesis with glucose or saccharose as carbon precursors and branched polyethylenimines (bPEI) to passivate the surface and confer positive charges. We have carried out a complete physical and spectroscopic characterization of these CDs. In addition, we evaluated the CD-dsRNA binding capacity. Cucumber plants were sprayed with CD-dsRNAs to test the capacity of the nanocomposite to enter the plant with respect to naked dsRNA. For that, we investigated by RT-qPCR the presence of dsRNAs in proximal and distant leaves from the point of application. RNAi was also investigated to compare the effect of CDs in increasing the amount of local and distal siRNA products derived from the dsRNAs that entered the leaf.

## 2. Results

Our synthesis approach based on solvo-hydrothermal reactions for the pyrolysis of carbon precursors (glucose and saccharose) allowed the obtention of carbon dots with different characteristics. The functionalization of the glucose and saccharose that have neutral charge, with branched polyethylenimines that conferred the cations, and consequently the net positive charge, resulted in adequate CDs after the synthesis. In preliminary investigations, we tested bPEI of different molecular weights. After their physicochemical characterization we selected only one size range, as well as reaction times and temperatures, discarding the rest either for showing lower fluorescence or lump formation after lyophilization (not shown). Thus, the carbon dots that we selected for in-depth analysis in this work were obtained with glucose (gCD) or saccharose (sCD) and the 2 kDa MW bPEI (Appendix A). We separated high-molecular-weight carbon dots using 0.22 μm filtration and the precursors and small CDs with the 1 kDa MWCO dialysis membrane. The eluted CDs were in the range of 1–10 nm and were discarded in the subsequent studies for their irregular size distribution (see below). Further investigation was carried out to study the properties of the carbon dots retained in the dialysis bags and their evaluation in RNAi applications.

### 2.1. Physicochemical Characteristics of the Nanoparticles

#### 2.1.1. Optical Properties of the CDs

The ultraviolet–visible (UV–Vis) absorption spectra showed differences because of the different carbon precursor, reaction times, temperatures, and carbon precursor:bPEI ratios (Appendix A). In all the cases, a maximum at 233 nm was observed, which is ascribed to the π-π* transition of C=C [34]. Lower absorption in the spectra was observed when the temperatures and reaction times decreased, indicating probable lower efficiency in the synthesis of carbon dots. The higher the ratio carbon precursor:bPEI, the higher the absorption was, showing a direct effect on CD synthesis efficiency. Another peak with a maximum at 366 nm resulted in the case of sCDs. In the case of the glucose-derived CDs, a shoulder between 300–350 nm in the absorbance could be observed. For the subsequent analyses, we selected the CDs obtained at 180 °C for 6 h in a 1:1 (w:w) ratio (carbon precursor:bPEI). These CDs were submitted to dialysis and the eluate and retained fractions were lyophilized and analyzed. The eluates differed in their absorption spectra because of the different molecular sizes (Figure 1A). Both the eluate and the membrane-retained CDs showed a similar color (dark yellow) and resulted in fluorescence under UV light (Figure 1A, inset). After the lyophilization, the fraction retained in the exclusion membranes was weighed and used for the dsRNA nanocomposites in solution. Fluorescence spectra were measured as a function of the excitation wavelengths and showed maximum values at 468 nm (light blue) for both the gCD and the sCD when excited at 350 nm (Figure 1B). Amplitude of the peaks depended on the excitation wavelengths, with the increase of the emissions ranging between 325–400 nm for both CDs. The analysis of the fluorescence signals for equal amounts of the nanoparticles in solution showed that the gCD were about 30% more intense than the sCD. The quantum yield (QY) of the CDs resulted in 1% for the sCD and 2% for the gCD.

#### 2.1.2. Structural and Morphological Properties of the CDs

Raman spectroscopy provided results that were difficult to interpret due to the high fluorescence of the samples (not shown). Therefore, the surface functional groups and chemical composition of the CDs were investigated by FT-IR and X-ray photoelectron spectroscopy (XPS). For both the gCD and sCD, FT-IR analysis showed in the high-frequency region an intense and broad signal corresponding to the O-H bond stretching vibration around 3435 cm^−1^ (Figure 1C). At lower energy, two weak signals appeared at 2926 and 2852 cm^−1^, compatible with C–H stretching vibrations with *sp^3^* character. The intense signal that appeared at 1629 cm^−1^ can be assigned to C=C bond stress vibrations with a certain degree of conjugation, although a contribution from C=O stress vibrations in amides could not be discarded. At 1499 and 1458 cm^−1^, two signals appeared that could come from bending vibrations of –CH_3_. The signals at 1384 and 1062 cm^−1^ could indicate the presence of C–OH and C–O–C groups, although the assignments in this region are difficult since vibrational modes of different functional groups that could be present in the sample overlap, e.g., CH_3_ and CH_2_ deformations, among others. The FT-IR analysis of the sCD was very similar to that obtained for the gCD (Figure 1C). In the 1500–1300 cm^−1^ region, bands were much more resolved, although the frequencies do not vary, probably due to a higher concentration of compounds dispersed in the KBr matrix. Stretching vibrations for O-H and C-H could correspond to peaks at 3435 and 2926/2920 cm^−1^, respectively. A N–H bending vibration could be resolved at 1630 cm^−1^ and other peaks at lower wavenumbers could correspond to C–N and C–O–C vibrations [39,40,41]. It is worth mentioning that the FT-IR spectra of both nanoparticles were very similar, with the only noticeable differences observed when focusing on the 1200–1000 cm^−1^ region, where for the sCD sample two signals appear to be resolved at 1160 and 1051 cm^−1^ (shadowed area in Figure 1C). These signals might correspond to C-O/C-O-C bending.

Regarding the XPS analysis, in both CDs three strong peaks appeared at binding energies of 283.2, 399.2, and 531.6 eV (Figure 2), which could be associated with the C1s, N1s, and O1s, respectively [32,41,42]. The deconvolution of the C1s spectra (Figure 2B,F) exhibited three peaks at 285.0, 286.2, and 287.8 eV. The binding energy at 285.0 eV could correspond to the graphitic structure (C–C/C=C), the peak at 286.2 eV probably corresponded to C–N/C–O and peaks at 287.8 eV are generally associated with O–C=O [32,43,44]. Deconvolution of the N1s spectrum displayed a peak at 399.4 eV (Figure 2C,G), that could correspond to amine or amide groups, and cannot be specifically resolved [43]. Finally, deconvolution of the C1s spectrum showed peaks at 531.3 and 532.6 eV which could correspond to C=O and C–O vibrations, respectively (Figure 2D,H). Besides, atomic concentrations were calculated (Appendix A) and the N/C ratios resulted in 21.9% and 22.6% for the sCD and gCD, respectively. Thus, both CDs seemed to be N-doped and exhibited hydrophilic groups on their surfaces.

The hydrodynamic diameters of the CDs were estimated using the Zetasizer, which determined that the particles retained in the dialysis bags averaged 5 nm for the gCDs and 4 nm for the sCDs (Figure 3A,C). The nanoparticles resulting from the eluate showed a less defined range of sizes (Figure 3B,D). TEM provided additional evidence for the size of the gCD nanoparticles present in the fraction retained in the dialysis membrane (Figure 4A). Inspection of the gCDs at higher resolution (Figure 4B) and Fourier transform analysis of the TEM images demonstrated that the gCDs had crystalline structure (Figure 4C).

On the other hand, the Zetasizer allowed the determination of the ζ potentials of the colloidal dispersions of the nanoparticles, averaging 9.54 and 9.92 mV for the sCD and gCD, respectively, indicating electrostatic positive charges for both nanoparticles (Figure 3E–H). These values were not very high and point out some instability in the colloidal dispersion. The electrophoretic mobilities in deionized water were similar, being 0.74 and 0.77 μm cm V^−1^ s^−1^, for the sCD and gCD, respectively, and the conductivities were 8.41 and 5.2 mS m^−1^ for the respective gCD and sCD nanoparticles. Finally, the isoelectric points were at pH 9.68 and 8.93, for the gCD and the sCD, respectively (Appendix A).

### 2.2. Characteristics of CD-dsRNA Nanocomposites

Binding of CDs and dsRNA was performed at room temperature and was quickly produced. The gCD-dsRNA and sCD-dsRNA nanocomposites showed an electrophoretic mobility of −1.56 and −1.12 μm cm V^−1^ s^−1^, respectively, and their corresponding ζ potentials were −7.7 and −14.0 mV, which could be compared with the −26.0 mV of pristine dsRNA in water (Figure 5). Hydrodynamic diameters of the nanocomposites were higher than the corresponding ones to the nanoparticles alone and resulted in differences in pristine dsRNA (Figure 5). The dsRNA molecules in water suspension showed a range of sizes, with a major peak at 45 nm and two other peaks at 1000 and 1400 nm, probably indicating different aggregation states of the molecules. Interestingly, when the nanoparticles were added, it resulted in single peaks for the nanocomposites of 350 and 160 nm diameter for the gCD-dsRNA and sCD-dsRNA, respectively. For further analyzing the CD-dsRNA interactions, we performed gel retardation assays. Increasing the CD:dsRNA ratio resulted in reducing the electrophoretic mobility to the positive electrode in agarose gels, indicating the progressive binding of the nanoparticles to dsRNA that increased the ζ potentials (Figure 6A,B). The fluorescence of the dsRNAs decreased as we increased the concentration of the CDs, plausibly because of the competition of the CDs with the RedSafe staining for intercalating in dsRNA molecules.

A nuclease protection assay determined that the nanoparticles at not saturating conditions did not protect the dsRNA for the action of the RNAse A (Figure 6C). Moreover, coated dsRNA degraded at lower nuclease concentration than pristine dsRNA. Conceivably, gCDs remain bound to degraded dsRNA fragments as their migration under the electrical current is reduced with respect to the RNAse-degraded naked dsRNA. Finally, FITC-labeled dsRNA could also bind to the CDs as shown by retarded migration in the electrophoresis of gCD-*dsRNA (Appendix A).

### 2.3. Detection of dsRNAs and siRNAs in Plants after Spraying Naked dsRNA or gCD-dsRNA

We prepared dsRNAs for spraying on cucumber leaves either naked or in the form of gCD-dsRNA nanocomposite. Typically, for the naked dsRNA and the nanocomposite we applied 3.5 μg of in vitro synthesized dsRNA per leaf (1X) (Appendix A). For the nanocomposite, the same amount of gCDs was added. In addition, 10-fold dilutions (0.1X) of the dsRNA and the gCD-dsRNA nanocomposites were prepared. Three days after the spraying, leaves were washed thoroughly with distilled water and once leaf surfaces were dry, we collected the samples for the analyses. Next, we performed RNA extractions for the quantitation of long (ds)RNAs and the siRNAs derived from them. Analysis of the Cq values allowed calculating the ΔCq (Cq_CP-dsRNA_-Cq_18S)_ for each sample in the different conditions (Figure 7A). Calculation of the ΔΔCq between the different dsRNA preparations showed that in the gCD-dsRNA(1X) samples, the amount of specific RNA exceeded 50-fold the amount present in the samples that were sprayed with naked dsRNA. When a 1:10 dilution of gCD-dsRNAs (0.1X) was sprayed on a group of plants and compared with a set sprayed with undiluted naked dsRNA, the amount detected in the plants was in the same order of magnitude (Table 1).

Regarding the siRNAs derived from the RNAi processing in the cell of the dsRNA, a comparison was performed (Figure 7B). For that, we quantified by RT-qPCR the 6125-vsiRNA that was previously identified by high-throughput sequencing [22]. Next, we observed that the 6125-vsiRNAs were 13.6-fold more abundant in the set of samples sprayed with gCD-dsRNA (1X) than with naked dsRNA (1X) (Table 1). Moreover, diluted gCD-dsRNA (0.1X) could produce in the leaves a similar siRNA amount to the undiluted dsRNA (1X).

Systemic movement was also investigated by the detection and quantification of the dsRNA and the 6125-vsiRNA in a distal leaf that was protected of the spraying with a cover foil (Appendix A). In this case, the (ds)RNA detected in the leaves was 1.2 × 10^3^-fold higher in plants sprayed with the gCD-dsRNAs with respect to naked dsRNA (Figure 7C; Table 1). With respect to the derived siRNAs (6125-vsiRNA), a consequence of the active RNAi machinery in the cells, they were also 12.4-fold more abundant in the distal leaves of plants sprayed with gCD-dsRNAs (Figure 7D). Remarkably, when comparing the dsRNA and vsiRNA in proximal and distal sites, it could be observed that the rates of distal versus local (ds)RNAs and vsiRNAs were two and one order of magnitude higher when the dsRNA was coated with the CDs than with naked dsRNA, suggesting that coated dsRNA improved long distance movement (Table 1).

### 2.4. Detection of gCD-Coated and Naked FITC-Labeled dsRNAs on Cucumber Plants Using Confocal Microscopy

To further investigate the capability of gCDs for enhancing the dsRNA entry, we used FITC-labeled dsRNA that was applied either naked or coated with the gCD onto cucumber plants by using the spraying. Samples were observed under the microscope before and after strong washing of the leaves with DD water (Figure 8). Strong fluorescence signals were observed in samples sprayed with naked or coated dsRNA*FITC, however, after the washing step, only weak signals could be observed in samples sprayed with gCD-dsRNA*FITC. Conversely, in samples sprayed with coated dsRNA*FITC, strong fluorescence signals remained, indicating that more dsRNA infiltrated consistently in the leaves. Furthermore, observations at a higher resolution showed that fluorescence signals mostly accumulated in the apoplast or in the cell walls (Figure 9). When the gCDs were used, the fluorescence signals appeared well distributed in the leaves (Figure 9C,D). In contrast, when the leaves were treated with naked dsRNA*FITC, the fluorescence signals appeared in patches (Figure 9A,B). The carbon dots could not be detected in confocal microscopy when illuminated at 405 nm or at lower wavelengths, probably because of their low fluorescence and quantum yield.

## 3. Discussion

In this work we obtained carbon nanoparticles by hydrothermal synthesis that present characteristics such as small size and positive charges, which made them useful as dsRNA carriers to deliver into the plant cell and elicit the RNAi machinery. Obtention of carbon dots using bench devices facilitates the synthesis and research on the possibilities of these particles in biological applications. CDs have been obtained by microwave pyrolysis using domestic microwave ovens using polyethylene glycol and distinct saccharides such as glucose or fructose as precursors followed or not by dialysis [43,45]. Carbon dots were also obtained by microwave pyrolysis using citric acid passivated with PEI for siRNA binding [32,46]. The same method was used to obtain CDs using citric acid passivated with ethylenediamine followed by dialysis [47]. Nevertheless, domestic microwave ovens vary in power, which makes it difficult to homogenize the synthesis protocol [26,35,40,48]. Standard chemistry-specific microwaves that allow the use of organic solvents and precise adjustments have been used to obtain carbon dots for plant RNAi applications [18], but their availability limits access to laboratories with fewer resources. Alternatively, hydrothermal synthesis seems to facilitate the standardization of protocols and led us to prefer this alternative method [49,50,51]. Solutions of citric acid and PEI have been subjected to pyrolysis in a Teflon-lined autoclave at 100 °C for 2h, obtaining purified CDs after the dialysis with 0.5 kDa cut-off membranes [43]. In another example, citric acid and ethylenediamine were used for CD synthesis using hydrothermal pyrolysis with a Teflon lined autoclave followed by dialysis [42]. Thus, elemental carbon sources and passivation with molecules conferring positive charges such as PEI or ethylenediamine have been successfully used for obtaining CDs [32,43]. The physicochemical characteristics of the CDs obtained in this work were comparable to those described in the literature.

For the study of hydrodynamic diameter of the CDs, in addition to direct visualization with the TEM, we used the dynamic light scattering in the Zetasizer. In both CDs, dispersion sizes averaging 4 nm and 5 nm were observed for the sCD and gCD, respectively. According to TEM, particle size of CDS from simple carbon sources vary by 3–12 nm [35,41,43,46]. Although we observed particles exhibiting well-resolved lattice fringes as reported in the literature, there are some other particles where that pattern was not visible, as has been reported before [41,46]. This is generally explained as the non-crystalline PEI chains that are wrapped around the crystalline part of the particles, making them undetectable. These small, fringe-free particles can be poorly visualized, and there is a possibility that our samples contain a proportion of these particles. Alternatively, it could be that some CDs were oriented along specific directions and with lattice planes large enough to be resolved by TEM, so that the fringes could be observed [25].

In the absorption spectra, the peak at 300–350 nm could be attributed to the presence of particles of different sizes and the distribution of the different surface energy traps of the carbon dots [52]. The fluorescence spectra from the gCDs and sCDs showed similar results to those of CDs reported in the literature that were obtained using simple compounds [35,43,53]. In these examples, there is a maximum emission value around 460–470 nm when the samples are excited at 350–360 nm, similar to our results for the gCD and sCD. We have observed how the position of the emission peak shifts from blue to green as the excitation wavelength increases from 350 nm to 500 nm. Therefore, our products showed fluorescence when illuminated with UV light, which was indicative of the presence of carbon dots, as none of the reaction precursors fluoresce when illuminated with UV light. The origin of the fluorescence of the carbon dots is still subject to debate, but the most accepted explanation accepts that when illuminated with ultraviolet light, electrons present in certain functional groups, such as C=O, and C=N, are excited to a higher energy state and emit fluorescence in a coordinated fashion as they decay from the valence state [28,36,40,54].

According to the FT-IR analyses, our CDs showed peaks corresponding to the vibrations of the bonds between the C, O and N elements in our sample, very similar to the results obtained in other CDs that used different carbon precursors and synthesis methods [37,49,50]. In the XPS, the appearance of N1s peaks indicates that the N elements successfully entered the carbon skeleton of both the sCDs and the gCDs. In the C1s, the surface areas of the bands differed between the sCDs and the gCDs, with the corresponding areas to the 285 eV in the gCDs lower than the corresponding ones to the sCDs. Therefore, a higher proportional rate of C–N/C–O bending was present in the sCDs. Besides, the FT-IR analysis agreed with the XPS in the description of the functional groups present in the surfaces of the CDs. On the other hand, the ζ potential, that was positive in both CDs, points to the passivation of the bPEI as shown by the presence of N covalently bonded on the surfaces of the CDs. The CDs obtained in this work showed low quantum yields, but not far from others used in cell labeling, that were around 3.5% [55].

CDs have been used in biomedicine for drug delivery (reviewed in [51]) and bioimaging [35,46,49]. When adequately passivated, they bind electrostatically to nucleic acids, and consequently have been used for NA delivery in living organisms [12,32,48,52,56,57,58]. Furthermore, CDs have been used to label DNA instead of commercial fluorophores [58,59]. When synthesizing this type of particle, we observed that they were adequate for our objective, as there was an effective binding between the nanoparticles and dsRNA. This was evident when dsRNA bands were delayed with respect to their corresponding position, either because the binding of the particles makes the molecule heavier, so it was expected to migrate less in the gel, or because the positive charges of the CDs offer resistance to migration towards the positive pole of the gel. Moreover, measurement of hydrodynamic diameters showed different sizes for the nanocomposites and their separate components. In another report, plasmid DNA (p-DNA) formed a complex with arginine and glucose-derived carbon dots obtained by microwave pyrolysis [48]. The CD-pDNA complex increased the diameter to 10–30 nm with respect to the 1–7 nm of the CDs, as determined by the Zetasizer. Our CD-dsRNA nanocomposites showed larger diameters, probably because of the linear nature of the dsRNA molecules versus the circular plasmid DNA. Chitosan and quaternary chitosan-derived CDs have been used to form complexes with dsRNA for shrimp virus control [60]. The nanocomposites varied in size between 350–650 nm and 150–350 nm, depending on the CD:dsRNA ratios. Progressive increase of the ζ potential was observed when increasing the CD:dsRNA ratio, being positive at ratios higher than 1.7:1 in chitosan-dsRNAs and 0.24:1 in quaternary chitosan CD-dsRNAs [60]. In our CD-dsRNA complexes, we have observed that ratios higher than 10:1 were positive. The gCDs-dsRNA used in this work for plant transfections were electronegative, as we used a 1:1 ratio in the composition. Wang and co-workers [32] obtained citric acid and PEI-derived CDs for siRNA binding. The hydrodynamic particle sizes of the CDs were 3.9 nm and 4.7 nm for the CD-siRNA complex, and the zetapotential was positive both for the CDs and the CD-siRNA used for human cell transfections. Binding to plasmid DNA has been achieved with microwave pyrolysis synthesized CDs derived from glycerol and PEI [61]. The CD-pDNA particles were 200 nm in size. CDs are reported to protect siRNAs from RNAse [18,61]. However, we did not observe protection on the dsRNAs by the gCDs in non-saturating conditions from RNAse A, suggesting that gCD-coated dsRNAs are fully accessible to plant RNAse III, effectively triggering the RNAi response and siRNA production. This assumption is supported by the abundant siRNAs detected in plants sprayed with gCD-dsRNA.

Regarding their application in plants, we have selected for our research the gCD nanoparticles for the obtention of the dsRNA nanocomposite formulations as they showed higher fluorescence and quantum yield, better solubility, and higher electrostatic charge than the sCDs. Once the spraying experiments were carried out, it could be observed how a three-magnitude order higher amount of dsRNA entered the plants when coated with the gCDs. Once in the plant, the dsRNA could elicit the RNAi machinery, resulting in its processing into siRNAs. The amount of siRNA produced was 50-fold higher in leaves sprayed with coated dsRNA. In distal leaves that were kept protected from the dsRNA spraying, we could detect both dsRNA and siRNAs, evidencing systemic movement. Remarkably, a three orders of magnitude higher dsRNA and one order of magnitude higher siRNA was detected in distal leaves of plants sprayed with coated dsRNA in comparison with naked dsRNA. Therefore, carbon dot coated dsRNAs may be effective in inducing local or systemic silencing, by requiring smaller amounts than naked dsRNA.

Although to our knowledge no other reports of quantitation of dsRNAs and vsiRNAs when applying CD-dsRNA have been reported, other authors describe more efficient siRNA release in plants when coated with CDs [18]. An increase in local and systemic GFP silencing has been reported when applying siRNA coated with carbon dots in *N. benthamiana* 16c, but only when surfactants were included in the formulations [18]. Alternatively, naked siRNAs produced systemic GFP silencing in *N. benthamiana* 16c when sprayed with high pressure on the leaves [12], to the point of damaging them. In our case, the pressure exerted when inoculating to obtain satisfactory results was medium, without damaging the leaves. Hence, the more effective local and systemic silencing of GFP expression observed in 16c plants when siRNA is delivered as CD-nanocomposites [18] can be explained in terms of higher amounts of siRNA available and the possible improvement of systemic movement of the coated RNAs, as resulting from our research. Although systemic GFP silencing has been so far observed only in *N. benthamiana* 16c plants [62], we and others have shown that virus-derived dsRNAs, or at least long RNA molecules, move systemically in the plant, and are subjected to the RNAi machinery producing siRNAs and lead to symptom reduction after virus inoculation [15,20,22]. Thus, improving delivery methods for siRNA/dsRNA in foliar application will either decrease the amount of dsRNA needed or improve its effectiveness. On the other hand, we have shown that free and CD-coated FITC-dsRNA can cross the cuticle of cucumber plants and accumulate primarily on the outside of the cell, the apoplast. Leaf washes showed that a large amount of naked dsRNA-FITC resulted in loss and therefore does not enter the plant as reported in *N. tabacum* [17]. LDH conjugates also improved internalization of plasmid DNA into human cells [29]. Therefore, nanoparticles, either LDHs or CDs, enhance the entry of dsRNA in the plant and reduce washing losses, increasing the efficiency of the topical spraying process. Further research is currently underway to investigate the movement to other growing parts of the plant and the stability over time of the sprayed CD-dsRNA. Recently, in a preprint report, it has been shown that unprocessed dsRNA molecules accumulate in the apoplast and move to distal parts of the plant [63]. Another important area of research is the improvement of the application of nanoparticles and their conjugates, via the development of formulations that in some cases can employ surfactants and other additives to improve the permeability of plant cell membranes [24]. However, in certain applications such as virus control, the use of permeabilizers can contribute to the opposite effect, i.e., favor virus movements within the plant (L. Ruiz, D. Janssen, pers. comm.). Nevertheless, another remaining question to address is the maximum amount of externally supplied dsRNA that a plant leaf can take and process into siRNA.

Finally, the use of carbon nanoparticles in agriculture is an object of concern because of their potential effects on plant development or contamination of the environment. Damages of carbon nanoparticles in the plants have been shown to be dosage dependent [64]. Thus, given the extremely small amounts (in the order of micrograms) that we apply to the plants to deliver dsRNA, we do not foresee problems in plant development. In preliminary applications of gCDs to cucumber plants for dsRNA control of cucumber green mild mosaic virus, we have not observed abnormal changes in plant growth (not shown). Therefore, regulatory agents should consider the limited amount of the active agents (dsRNA/siRNAs) and companion adjuvants in the formulations that will probably be required in agricultural applications in the field. In conclusion, our analyses have shown that we have obtained carbon nanoparticles using hydrothermal synthesis and dialysis and which are adequate for their use in RNAi applications in plants, such as crop protection against viruses, fungi, or insects.

## 4. Materials and Methods

Materials: Glucose (G8270) and the 800 Da branched polyethylenimine (408719) were purchased from Sigma-Aldrich (St. Louis, MO, USA). Saccharose (131621) was purchased from Panreac (Barcelona, Spain). The 2000 Da branched polyethylenimine (06089) was purchased from Polysciences (Warrington, PA, USA) and the 5000 Da branched polyethylenimine (Lupasol G100) was purchased from BASF (Ludwigshafen, Germany). The seeds of cucumber (*Cucumis sativus* cv. “Bellpuig”) were purchased from Semillas Fitó (Barcelona, Spain).

### 4.1. Synthesis of Carbon Dots

Briefly, 2 g of glucose or saccharose were dissolved by strong stirring in 10 mL of MilliQ water containing 2 g of branched polyethylenimines (bPEI) of 800, 2000, or 5000 Da molecular weight. Next, the solution was transferred into a stainless-steel autoclave with a Teflon liner of 100 mL capacity and heated at 120–180 °C for 4–6 h. After cooling to room temperature, the resulting dark yellow solutions were filtered through a 0.22 μm filter (Millipore, Merck, Darmstadt, Germany). The filtrates (10 mL) were then allocated in 1000 Da molecular weight cut-off (MWCO) dialysis bags (Spectra/Por, Fisher, MA, USA) sealed with claps and dialyzed against 40 mL of MilliQ water for 24 h in a 50 mL Falcon tube in agitation. The eluate was recovered, lyophilized, and kept apart. Next, the bag was moved to a recipient with 2000 mL of DD water and the dialysis was done after stirring for 24 h. After that, the water in the recipient was removed and changed for another 2000 mL of DD water and the dialysis continued for another 6 h. The content of the dialysis bags was then recovered, lyophilized, weighed, and used for the subsequent analyses.

### 4.2. Characterization of the CDs

The UV–Vis absorption spectrum of the CDs was recorded using the Multiskan GO microplate spectrophotometer (ThermoFisher, MA, USA). The fluorescence measurements were performed using the FLS920 Spectro fluorophotometer (Edinburgh Instruments, Livingston, UK) with a slit width of 2.5 nm for both excitation and emission. The quantum yields, QY (Φ), were calculated with the 1-M-1 integrating sphere in the same equipment. The morphology and size of the CDs were examined using TEM with the FEI Talos F200X microscope (ThermoFisher). The hydrodynamic diameter of the particles in MilliQ water were determined using the Zetasizer (Zetasizer Nano ZS, Malvern, UK) using dynamic light scattering. The electrophoretic mobility of the particles was determined using phase analysis light scattering using the same instrument. The Zeta (ζ) potential was next derived from the electrophoretic mobility using the Hückel approximation.

Fourier transform infrared (FT-IR) spectra were collected using the Tensor 27 spectrophotometer (Bruker, Bremen, Germany) using a Gate Single Reflection Diamond ATR System accessory. A standard spectral resolution of 4 cm^−1^ in the spectral range 4000–400 cm^−1^ and 64 accumulations were used to acquire the spectra. X-ray photoelectron spectroscopic (XPS) analyses were performed on a Multilab System 2000 X-ray photoelectron spectrometer (ThermoFisher). Raman spectroscopy was done with the Raman Spectrometer-Microscope NRS 5100 (Jasco, Tokyo, Japan). Excitation for Raman measurements was carried out by Nd:YAG laser with wavelengths of 325, 532 or 785 nm. For detection, the device included a thermoelectrically cooled CCD (Charge Couple Device) camera.

### 4.3. In Vitro Synthesis of dsRNA

Plasmid L4440gtwy, a derivative of L4440 that carries a double T7 promoter at both sides of the Gateway attR1/attR2 cloning sites was a gift from G. Caldwell (Addgene plasmid # 11344; http://n2t.net/addgene:11344; RRID: Addgene_11344) and was kept in *Escherichia coli* DB3.1. Plasmid pL4440-CP resulted from the Gateway cloning of a 464 bp segment of the coat protein (*cp*) gene of cucumber green mild mottle virus (CGMMV) [65]. The plasmid was used to transform *E. coli* strains Top10. Next, *E. coli* cells that contained pL4440-CP were grown in LB supplemented with carbenicillin (100 μg/mL) followed by plasmid DNA extraction with the High Pure Plasmid Isolation Kit (Sigma). For the in vitro synthesis, we linearized plasmid pL4440-CP in independent reactions with *Bgl*II and *Hind*III (NEB). Once linearized, the plasmid was purified and used as a template in a single reaction for dsRNA synthesis using the HiScribe T7 High Yield RNA synthesis kit (NEB). For plasmidic DNA removal, the synthetic dsRNAs were treated with DNAse I (Sigma) for 10 min at 37 °C and recovered by precipitation. After the synthesis, the dsRNA was heated at 85 °C and allowed to cool at room temperature. DsRNA quantitation was done with the ND-1000 spectrophotometer (Nanodrop, Wilmington, DE, USA) and examined in 2% agarose gels stained with RedSafe (Intron, Seongnam, Korea) under UV light.

### 4.4. Characterization of the CD-dsRNA Nanocomposites

The CDs were resuspended in MilliQ water and used to bind the dsRNA in water at room temperature. To characterize the interaction of the dsRNA and the CDs in the nanocomposites we used several approaches. The Zetasizer was used to measure the size, charge, and electrophoretic mobility of the nanocomposites. Electrophoresis allowed the analysis of the migration in 2% agarose gels of the dsRNA, the CDs, and the nanocomposites. Nuclease protection assays were performed with RNAse A (Sigma) at 37 °C followed by gel electrophoresis.

### 4.5. Application of dsRNA to Cucumber Plants

The cucumber seeds were sown after a preliminary soaking for 6 h and transferred to pots in a growth room. When the seedlings had 2 fully expanded leaves, they were sprayed with naked dsRNA or CD-dsRNA using an artist airbrush at 2.5 bar. In each spraying, we used 3.5 μg of dsRNA or dsRNA combined with the CD in 1:1 (w:w) proportions (dsRNA 1X, gCD-dsRNA 1X). In other samples we used 1/10 diluted dsRNA or 1/10 diluted CD-dsRNA (dsRNA 0.1X, gCD-dsRNA 0.1X). These preparations were applied on 4 cm^2^ in a leaf from each of the six plants (biological replicas) that were used in the assays. Plants were kept in the growth room at 25 °C and 16 h/8 h light/dark cycles. At 3 days post application (dpt), two points of the plant were sampled, the site where the spraying was applied (point 1, Appendix A) and a distal leaf (point 2) that was previously covered with a foil to prevent the aerial arrival of the dsRNA.

### 4.6. Long-(ds)RNA Quantitation in Proximal and Distal Parts of the Plants

Three days after the dsRNA treatments (dpt), a circular hole punch was used to obtain approximately 100 mg of leaf tissue for the total RNAs extractions performed with the Trizol method. Six samples (biological replicates) in each condition were used for the analyses. The experiment was repeated twice and the results are from the average of the combined data. Total RNAs were quantified using the NanoDrop ND-1000 (Thermo Fisher Scientific, Waltham, MA, USA). For the cDNA obtention, we used 2 μg of the total RNA extract from each sample and the High-Capacity cDNA Reverse Transcription Kit (Applied Biosystems) using 10 pmol random nonamers (Takara, Shiga, Japan) in 20 μL reaction volume and according to manufacturers’ instructions. Each qPCR reaction (20 µL final volume), in triplicate, contained 1 μL of the cDNA, 10 µL of KAPA SYBR Green qPCR mix (KAPA Biosystems, Wilmington, MA, USA), and 500 nM each of primer CP197F (5′-TACGCTTTCCTCAACGGTCC-3′) and CP305R (5′-GCGTCGGATTGCTAGGATCT-3′). In separate reactions, we included the primers for the *C. sativus* 18S rRNA as a reference. Specificity of the amplicons obtained was checked with the Bio-Rad Optical System Software v.2.1 by means of melting-curve analyses (60 s at 95 °C and 60 s at 55 °C), followed by fluorescence measurements (from 55–95 °C, with increments by 0.5 °C). The geometric mean of their expression ratios was used as the normalization factor in all samples for measuring the quantification cycle (Cq). The relative expressions of the (ds)RNA amounts were compared based on the calculations done with the 2^−ΔΔCq^ method.

### 4.7. Quantitation of siRNAs in Proximal and Distal Parts of the Plant

Detection and quantitation by RT-qPCR of the small RNAs derived from the dsRNAs was performed as described previously [22]. In this section, we quantified the 6125-vsiRNAs derived from the sprayed dsRNA, as a product of the RNAi machinery from the plant cell. Briefly, the RNA extracted from the plant samples was polyadenylated with the poly A polymerase (NEB, Ipswich, MA, USA) and reverse transcribed using the primer polyT (5′-GCGAGCACAGAATTAATACGACTCACTATAGGTTTTTTTTTTTTVN-3′), as described by Shi and Chiang [66] and the High-Capacity cDNA Reverse Transcription Kit. The cDNA was then used to detect by qPCR the 6125-vsiRNA using the primer CG-6125 (5′- GCTAGGGCTGAGATAGATAATT-3′) and the universal reverse primer (URP) (5′-GCGAGCACAGAATTAATACGAC-3′). Reaction and cycling conditions were described previously [22]. For the reference with an endogenous plant siRNA, we used the primer CUC5.8S based on the 5.8S rRNA of *C. sativus* (5′-CTTGGTGTGAATTGCAGGATC-3′) [22]. Six biological replicas were included in each condition and the experiment was repeated twice. Each qPCR (technical repetition), including those for the 5.8S as internal control, was repeated three times. The specificity of the amplicons obtained was checked as above and the relative expressions of the vsiRNAs were calculated as described previously.

### 4.8. FITC-Labeling of dsRNA and Confocal Microscopy of Cucumber Leaves

For dsRNA labeling with the fluorochrome, we followed the same dsRNA synthesis method described above but including fluorescein-X-(5-aminoallyl)-UTP (Jena Bioscience) in the reaction mix following the manufacturer’s instructions. Cucumber leaves treated with FITC-dsRNA were observed by laser scanning confocal microscopy (SP5 II, Leica). To observe the FITC signals, the excitation laser was set to 488 nm and the detection filter was set to 520 nm.

## 5. Conclusions

Carbon dots obtained from glucose and saccharose passivated with PEI using a solvothermal method showed positive charges and showed similar properties to other CDs described in the literature. These CDs were capable of binding to dsRNA producing nanocomposites. When applied on plant leaves by spraying, the nanocomposites favored 50-fold the entry of dsRNA with respect to naked dsRNA. The dsRNA from the nanocomposite was detected in the opposite leaf (distal part of the plant) at >1000-fold with respect to naked dsRNA. Small interference RNAs, derived from the dsRNA, could be detected either from the nanocomposite or from naked dsRNA. The siRNAs derived from the nanocomposite were >10-fold more abundant with respect to the naked dsRNA. Moreover, in distal leaves, the siRNAs derived from the nanocomposite were 10-fold more abundant with respect to the derived from the application of naked dsRNA. DsRNA in the form of nanocomposite was shown to enter the plant with higher efficiency than naked dsRNA, as shown using FITC-labelled dsRNA and washing of the leaves. Labelled dsRNA accumulated mainly in the apoplast of the plant cell.

## Figures and Tables

**Figure 1 ijms-23-05338-f001:**
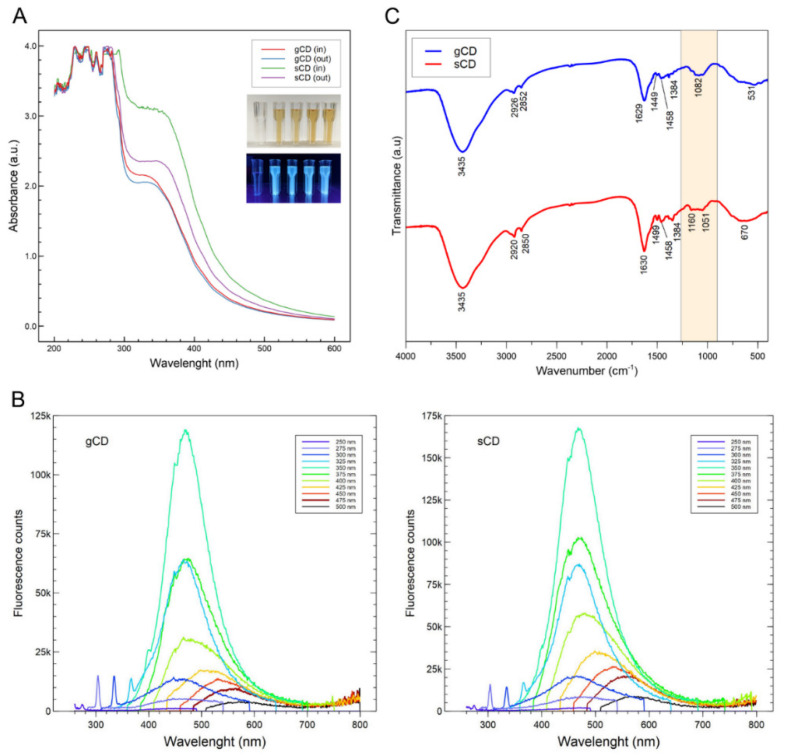
(**A**) Absorption spectra of the glucose (gCD) and saccharose (sCD) carbon dots obtained in this work. Aspect of the water dilutions of the nanoparticles when illuminated with white (upper inset) and ultraviolet light (lower inset); from left to right: solution of glucose and bPEI in water; gCD(in): gCD retained in the 1 KDa MWCO dialysis membrane; gCD(out): gCD eluate from the dialysis membrane; sCD(in): sCD retained and sCD(out): sCD eluate. (**B**) Wavelength-dependent emission spectra of the gCD and sCD carbon dots retained in the dialysis membranes. Wavelength scanning was performed from 250 to 475 nm with steps of 25 nm. (**C**) FT-IR spectra of the carbon dots retained in the dialysis membranes.

**Figure 2 ijms-23-05338-f002:**
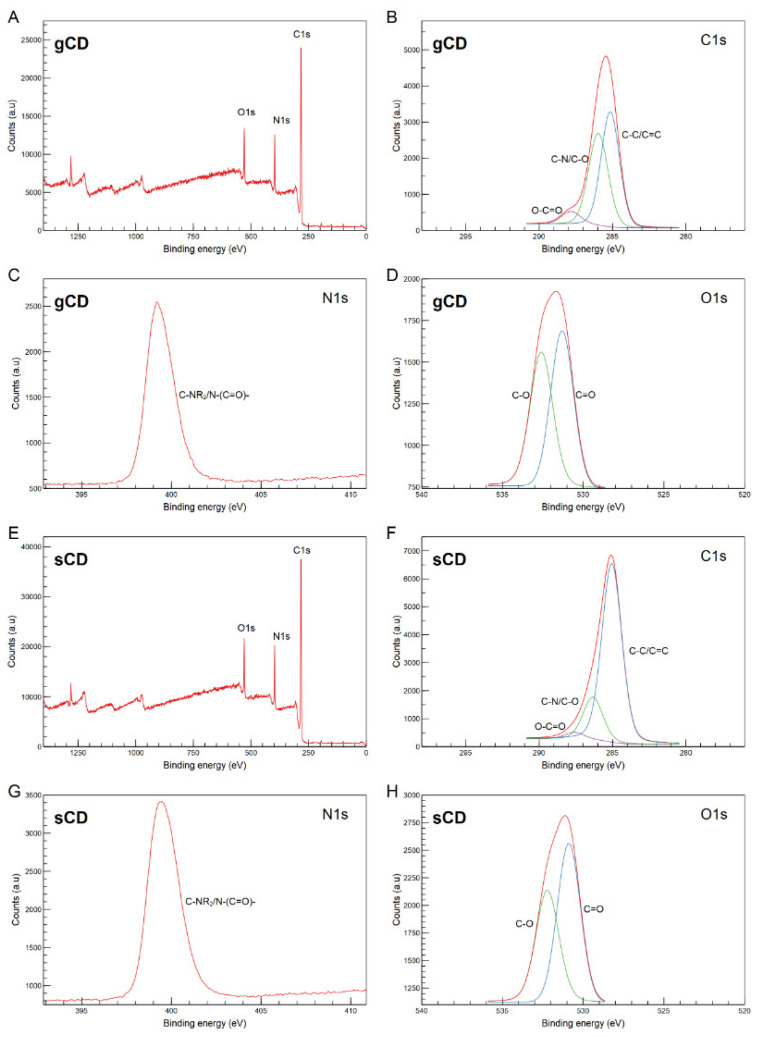
XPS spectra of the respective gCD and sCD. Survey spectra (**A**,**E**), and high-resolution deconvoluted spectra for C1s regions (**B**,**F**), N1s regions, and (**C**,**G**) and O1s regions (**D**,**H**).

**Figure 3 ijms-23-05338-f003:**
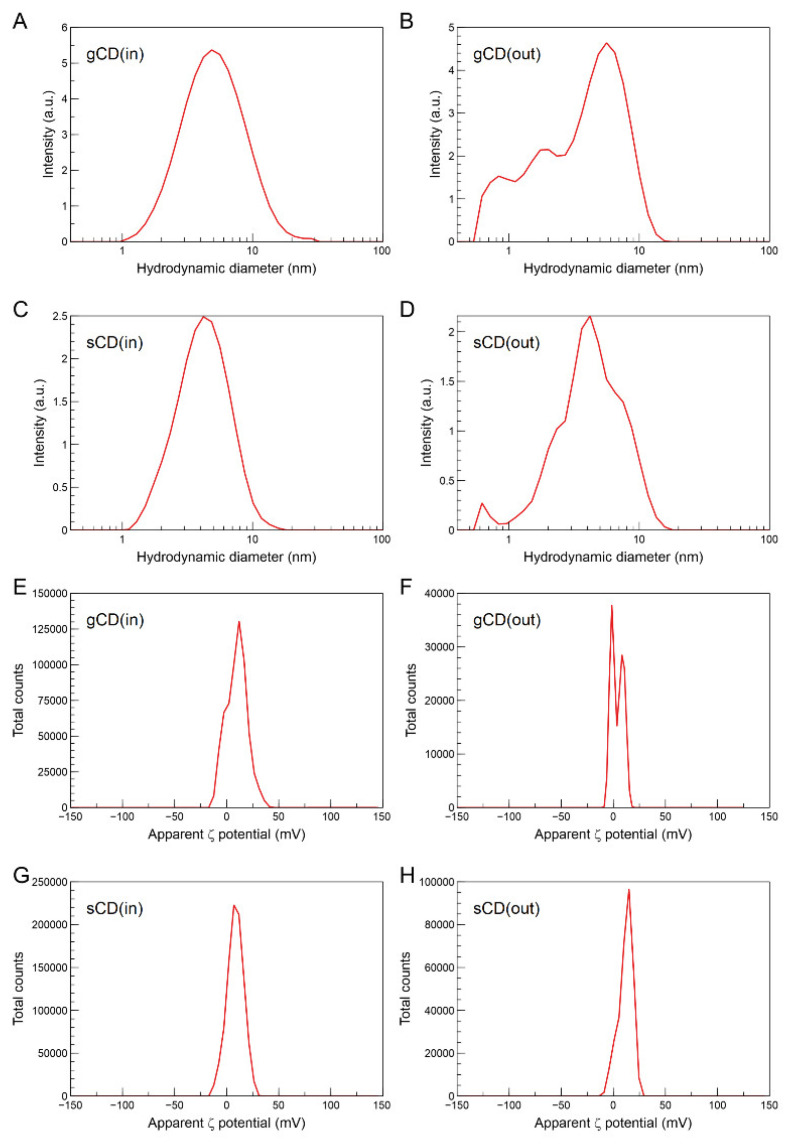
Hydrodynamic diameters of the carbon dots in water suspension according to the Zetasizer (**A**–**D**). gCD(in), gCD retained in the dialysis membranes; gCD(out), gCD) eluate; sCD(in), sCSs retained and sCD(out), sCD eluate from the dialysis membranes. Apparent ζ potential distributions of the retained and eluate gCDs and sCDs (**E**–**H**). The CDs were prepared using glucose or saccharose passivated with bPEI 2 KDa.

**Figure 4 ijms-23-05338-f004:**
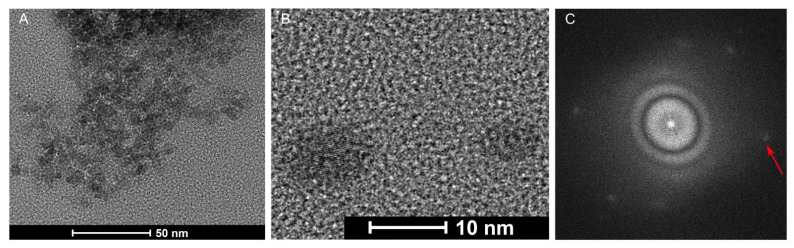
Transmission electron microscopy (TEM) images of gCDs. (**A**) TEM at 50 nm scale; (**B**) TEM at 10 nm scale allowing the observation of lattice fringes in the nanoparticles; (**C**) Fourier transform of the particles observed in image (**A**). The red arrow points to the white dots in the Fourier transform that indicate a crystalline structure.

**Figure 5 ijms-23-05338-f005:**
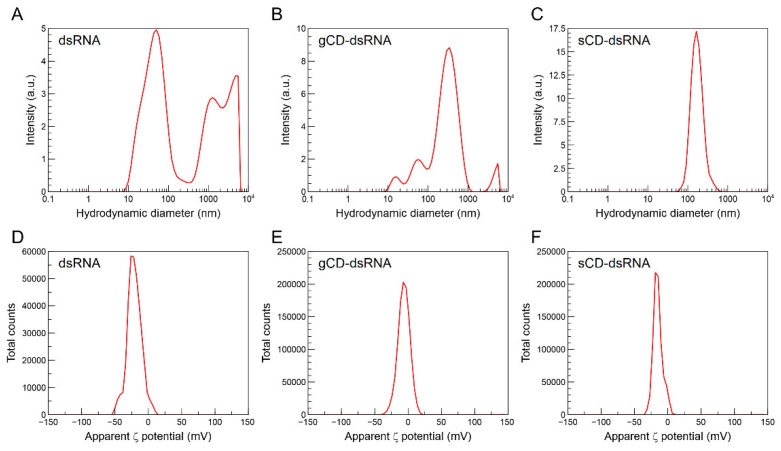
Distribution of hydrodynamic diameters (**A**–**C**) and apparent ζ potential distributions (**D**–**F**) and for the pristine dsRNA and the gCD-dsRNA and sCD-dsRNA nanocomposites.

**Figure 6 ijms-23-05338-f006:**
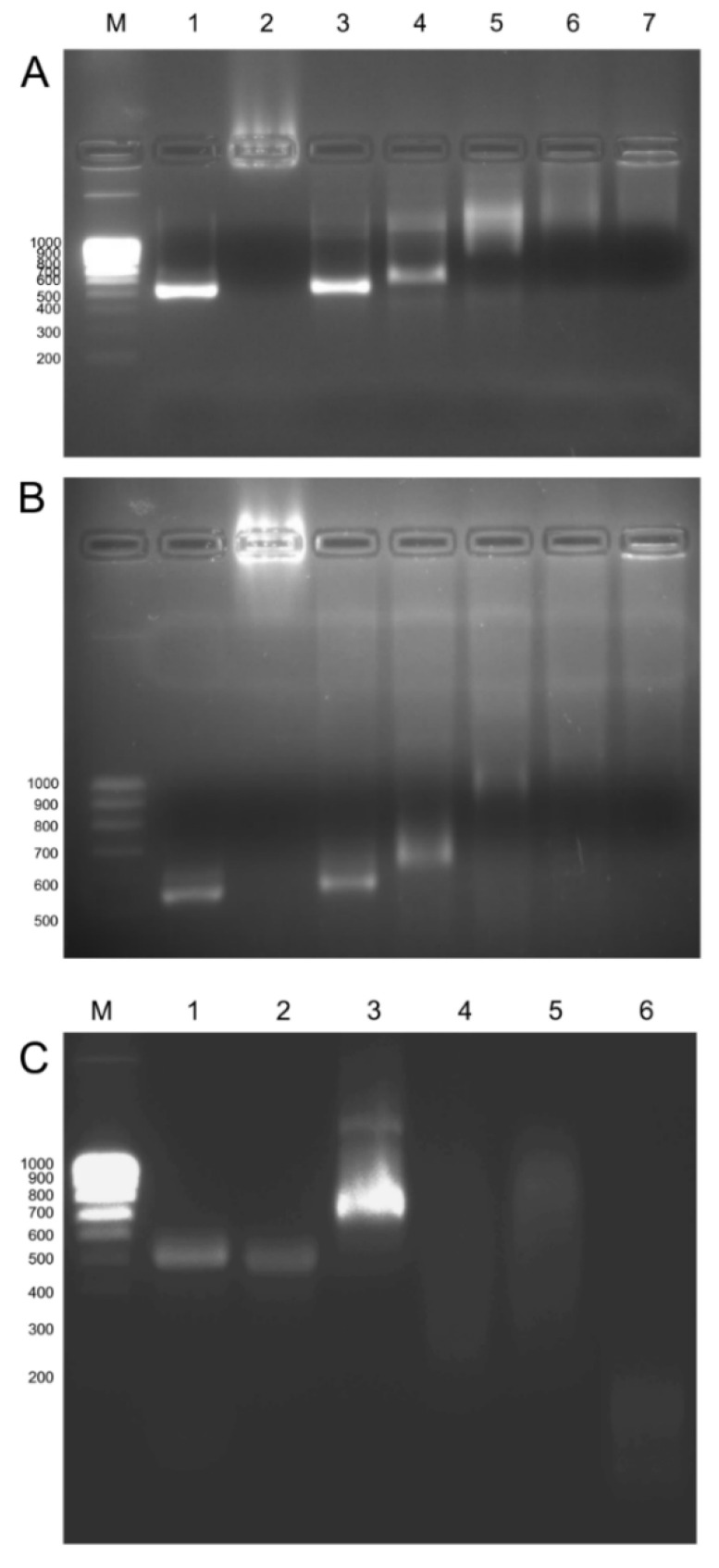
Delay of dsRNAs migration in 2% agarose gel electrophoresis when coated with increasing amounts of gCD (**A**) and sCD (**B**). (1) dsRNA; (2) gCD/sCD; (3) gCD/sCD:dsRNA [1:10]; (4) gCD/sCD:dsRNA [1:5]; (5) gCD/sCD:dsRNA [1:2.5]; (6) gCD/sCD:dsRNA [1:2]; (7) gCD/sCD:dsRNA [1:1]. (**C**) RNAse protection assays: (1) pristine dsRNA, (2) dsRNA and 0.125 U RNAse A 5 min, (3) gCD-dsRNA, (4) gCD-dsRNA and 1.25 U RNAse A 5 min, (5) gCD-dsRNA and 0.125 U RNAse A 5 min, (6) dsRNA and 1.125 RNAse A 5 min. M: NZY Tech Ladder V molecular weight marker.

**Figure 7 ijms-23-05338-f007:**
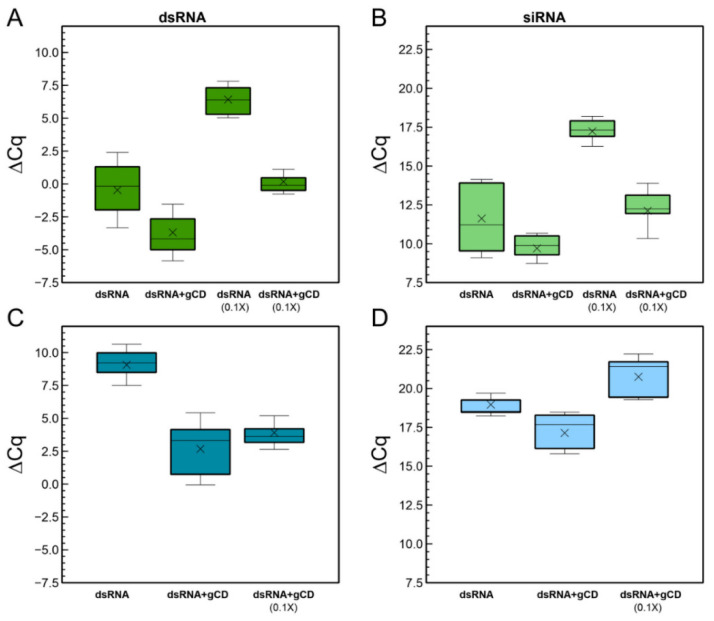
Quantitation of (ds)RNA in leaves after 3 dpt of the application of the gCD-dsRNA or naked dsRNA in local (**A**) and distal leaves (**C**). Quantitation of the derived 6125-vsiRNA in local samples (**B**) and distal leaves (**D**). Each boxplot shows the median (horizontal line), first and third quartiles (lower and upper limits of boxes), and the minimum and maximum values (delimited by the external whiskers). Mean values are indicated by crosses.

**Figure 8 ijms-23-05338-f008:**
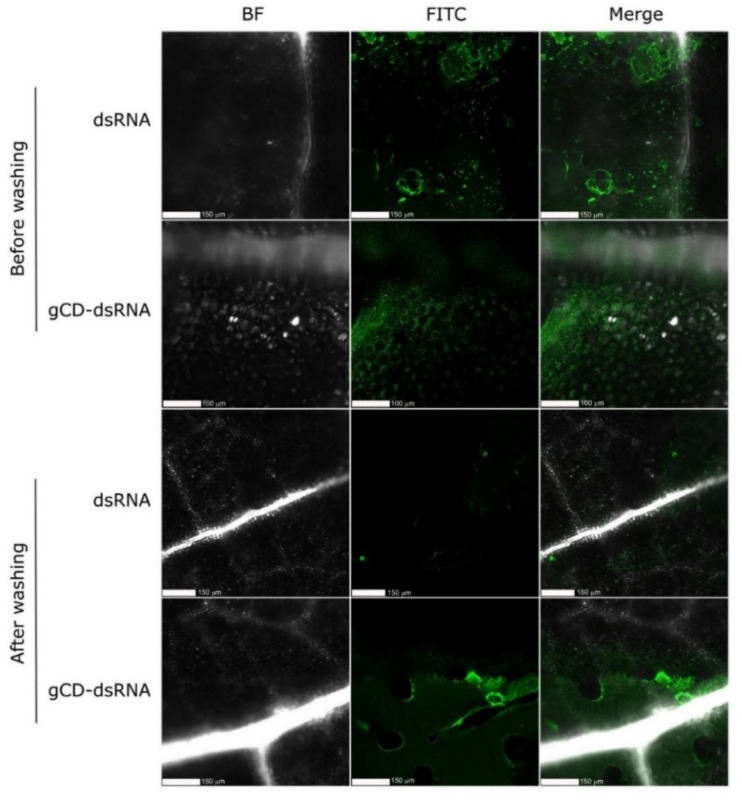
Confocal microscopy of cucumber leaves treated with dsRNA*FITC and gCD-dsRNA*FITC before and after washing them with distilled water. Column 1: bright-field (BF) images; column 2: FITC fluorescence images; column 3: BF and FITC merged images.

**Figure 9 ijms-23-05338-f009:**
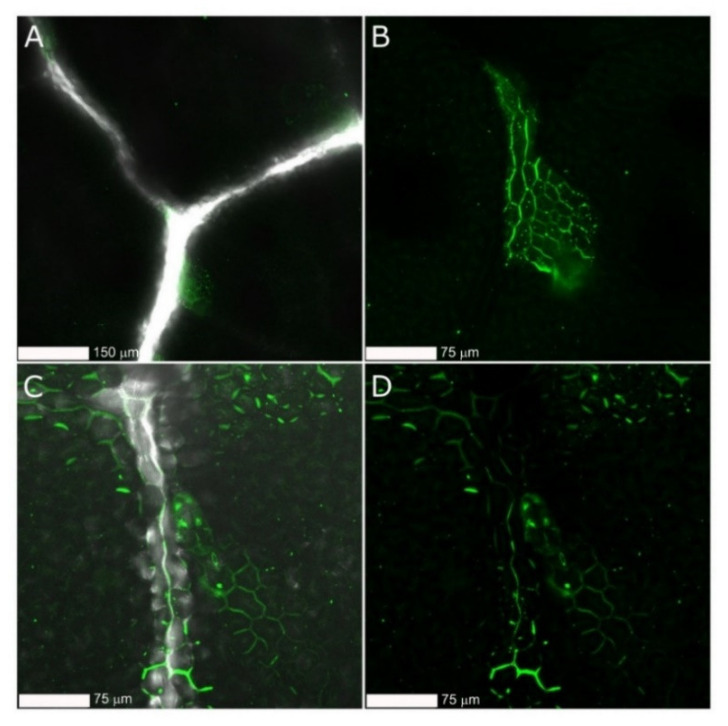
Confocal microscopy at higher resolution of cucumber leaves treated with dsRNA*FITC (**A**,**C**) and gCD-dsRNA*FITC (**B**,**D**) after washing the leaves with water.

**Table 1 ijms-23-05338-t001:** Comparisons of fold changes in the quantitation of (ds)RNA and 6125-vsiRNA in the site of application and on distal leaves after spraying dsRNA, 0.1X dsRNA, gCD-dsRNA or 0.1X gCD-dsRNA.

Condition	Comparison	Fold Increase	
		(ds)RNA	vsiRNA
Local leaves	gCD-dsRNA 1X vs. dsRNA 1X	50.4	13.6
	dsRNA 1X vs. dsRNA 0.1X	207.4	547.3
	gCD-dsRNA 1X vs. gCD-dsRNA 0.1X	95.7	22.4
	gCD-dsRNA 0.1X vs. dsRNA 1X	1.05	3.3
Distal leaves	gCD-dsRNA 1X vs. dsRNA 1X	1188.5	12.4
	gCD-dsRNA 1X vs. gCD-dsRNA 0.1X	6.89	74.2
	gCD-dsRNA 0.1X vs. dsRNA 1X	345.0	7.4
gCD-dsRNA 1X	Local vs. distal	1.7 × 10^3^	3.4 × 10^3^
dsRNA 1X	Local vs. distal	2.59 × 10^5^	3.3 × 10^4^

## Data Availability

Data available on request from the authors.

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
