# Peer review of "Carbon Dots Boost dsRNA Delivery in Plants and Increase Local and Systemic siRNA Production"

_ijms, 2022, doi:10.3390/ijms23105338_

Round 1

Reviewer 1 Report

While the topic is timely and of interest and the science seems sound there are serious issues that must be addressed before interpretation is achievable.

Major issues:

  1. Figure Legends are inadequate and do not allow interpretation
  2. Supplemental legends are inadequate and do not allow interpretation

Minor issues:

Line 10: missing comma after “work”

Line 14: missing comma after “mobility” and possibly one needed before “and were suitable”

Line 19: missing a word between “leaves” and “showed”

Line 22: “in the plant” I think would work best as “into the plant”

Line 34: missing an article before “template”

Line 34: missing comma after “plants

Lines 40-43: the sentence spanning these lines is confusing, I think it needs some commas and possible subject-verb agreement fixes

Line 45: missing comma after “tRNAs”

Line 48: missing comma after “stability”

Line 50: missing comma after “virus”

Line 50: missing comma after “cases”

Line 51: I think it’s missing an article before “aqueous phase”

Line 54: Missing comma after “general”

Lines 57-60: this sentence does not make sense to me

Line 61: change “fundamentally” to “fundamental”

Line 63: missing comma after “pathogenic organisms”

Line 64: add commas around appositive “due to their nanometric scale”

Line 65: missing comma after “surface”

Line 73: change “of cardon dots” to “from carbon dots”

Line 75-76: add commas around appositive “once they are applied to the plant”

Line 80: change “exceeds” to “exceed”

Line 85: missing comma after “index”

Line 85: add “in” after “result”

Line 86: change “referred as” to “referred to as”

Line 88: change “consists in” to “consists of”

Line 93: missing comma after “ethylenediamine”

Line 99: missing comma after “chemical stability”

Line 105: missing comma after “biosensors”

Line 107: missing comma after “systemic movement”

Line 108: missing comma after “characterizing”

Line 108: missing hyphen in “glucose derived”

Line 124-128: sentence spanning these lines does not make sense

Line 139: missing comma after “temperatures”

Line 141: missing comma after “bPEI”

Line 146: missing an article before “maximum”

Lines 146-148: sentence is confusing

Line 148: missing commas after “analysis”

Line 152: I think it’s missing a hyphen in “membrane-retained” or the sentence is just confusing

Line 152: change “resulted” to “resulted in” and maybe “florescent” to “florescence”

Line 153: missing comma after “lyophilization”

Line 154: change “were” to “was” and “weighted” to “weighed”

Line 156: get rid of “the” before “both”

Line 160: change “more intense that” to “more intense than”

Line 161: add “in” after “resulted”

Line 166: missing comma after “sCD-IN”

Line 175: “high frequency” should have a hyphen

Lines 184-185: “such as, for example” is redundant

Line 198: missing comma after “N1s regions”

Line 200: missing comma after “399.2”

Line 202: missing coma after “286.2”

Line 206: change “that could” to “which could”

Line 209: add “in” after “resulted”

Line 216: change “that determined” to “which determined”

Line 230: change “point out to some” to “point out some”

Line 244: change “resulted different to” to “resulted in differences to”

Line 248: change “160 nm of diameter” to “160 nm diameter”

Line 305: change “prevented of spraying” to either “protected from spraying” or “prevented by spraying” (not sure what the sentence is trying to say

Line 308: add article in front of “consequence”

Line 320: is “microcopy” supposed to be “microscopy”?

Lines 333/334: is “microcopy” supposed to be “microscopy”?

Line 337: missing article in front of “higher resolution”

Line 354: change “made us to prefer” to “led us to prefer” or something similar

Line 356: add hyphen in “Teflon lined”

Line 372-374: sentence here doesn’t make sense, I think it’s missing a verb

Line 377: change “resulted similar to” to “had similar results to”

Lines 382-384: reword sentence, potential rewording- “Therefore, our products showed fluorescence when illuminated with UV light, which was indicative of the presence of carbon dots, as none of the reaction precursors fluoresce when illuminated with UV light.”

Line 391: missing comma after “O”

Line 393: correct subject-verb agreement by changing “indicate” to “indicates”

Line 399: correct subject-verb agreement by changing “point out” to “points out”

Line 401: “The quantum yield of the CDs obtained in this work resulted low” doesn’t make sense

Line 402: add “that” after “reported”

Line 407: correct subject-verb agreement by changing “has” to “have”

Line 408: change “they resulted suitable” to “they had suitable results”

Line 420: add hyphen to “chitosan derived”

Line 420: correct subject-verb agreement by changing “has” to “have”

Line 428: change “resulted” to “were”

Line 440: missing comma after “better solubility”

Lines 443-444: The last half of the sentence seems to not be complete, maybe missing a verb?

Line 445: “prevented to reaching the spraying” doesn’t make sense

Line 461: I think “higher amount” should be “higher amounts”

Line 465: correct subject-verb agreement by changing “moves” to “move”

Line 471: change “resulted lost” to “resulted in loss”

Line 485: change “external” to “externally”

Line 486: change “is being object” to “is an object”

Line 488: correct subject-verb agreement by changing “has” to “have”

Line 491: I think “cucumber plant” should be plural

Line 498: missing comma after “fungi”

Line 508: missing comma after “2,000”

Line 511: change “were flowed through” to “were filtered through”

Line 515: missing comma after “lyophilized”

Line 519: change “weighted” to “weighed” and add comma after

Line 552: add an article in front of “template”

Line 563: missing comma after “charge”

Line 564: missing comma after “CDs”

Line 577: confused by the phrase “spraying was made”

Line 585: missing comma after “obtention”

Line 600: change “derived of” to “derived from”

Line 601: missing comma after “in this section”

Line 618: missing comma after “fluorochrome”

Line 620: missing an article before “manufacturer’s”

Line 626: change “resulted similar to” to “had similar results to”

Line 635: “with respect to the derived from naked dsRNA” doesn’t make sense

Author Response

(Many thanks for the careful revision and the suggestions: we include our comments in blue)

REVIEWER #1

While the topic is timely and of interest and the science seems sound there are serious issues that must be addressed before interpretation is achievable.

Major issues:

Figure Legends are inadequate and do not allow interpretation

Supplemental legends are inadequate and do not allow interpretation

We have included more details and made several amendments to the figures, including the supplementary material, to help with their interpretation.

Minor issues:

Line 10: missing comma after “work”

Corrected

Line 14: missing comma after “mobility” and possibly one needed before “and were suitable”

Corrected

Line 19: missing a word between “leaves” and “showed”

Corrected “and”

Line 22: “in the plant” I think would work best as “into the plant”

Corrected

Line 34: missing an article before “template”

Corrected

Line 34: missing comma after “plants

Corrected

Lines 40-43: the sentence spanning these lines is confusing, I think it needs some commas and possible subject-verb agreement fixes

The sentence has been rewritten to improve its understanding.

Line 45: missing comma after “tRNAs”

Corrected

Line 48: missing comma after “stability”

Corrected

Line 50: missing comma after “virus”

Corrected

Line 50: missing comma after “cases”

Corrected

Line 51: I think it’s missing an article before “aqueous phase”

Corrected

Line 54: Missing comma after “general”

Corrected

Lines 57-60: this sentence does not make sense to me

This sentence has been rewritten.

Line 61: change “fundamentally” to “fundamental”

Corrected

Line 63: missing comma after “pathogenic organisms”

Corrected

Line 64: add commas around appositive “due to their nanometric scale”

Corrected

Line 65: missing comma after “surface”

Corrected

Line 73: change “of cardon dots” to “from carbon dots”

Corrected

Line 75-76: add commas around appositive “once they are applied to the plant”

Corrected

Line 80: change “exceeds” to “exceed”

Corrected

Line 85: missing comma after “index”

Corrected

Line 85: add “in” after “result”

Corrected

Line 86: change “referred as” to “referred to as”

Corrected

Line 88: change “consists in” to “consists of”

Corrected

Line 93: missing comma after “ethylenediamine”

Corrected

Line 99: missing comma after “chemical stability”

Corrected

Line 105: missing comma after “biosensors”

Corrected

Line 107: missing comma after “systemic movement”

Corrected

Line 108: missing comma after “characterizing”

Corrected

Line 108: missing hyphen in “glucose derived”

Corrected

Line 124-128: sentence spanning these lines does not make sense

The sentence has been fully rewritten.

Line 139: missing comma after “temperatures”

Corrected

Line 141: missing comma after “bPEI”

Corrected

Line 146: missing an article before “maximum”

Added “a”

Lines 146-148: sentence is confusing

This sentence was rephrased.

Line 148: missing commas after “analysis”

Corrected

Line 152: I think it’s missing a hyphen in “membrane-retained” or the sentence is just confusing

Added hyphen

Line 152: change “resulted” to “resulted in” and maybe “florescent” to “florescence”

Corrected

Line 153: missing comma after “lyophilization”

Corrected

Line 154: change “were” to “was” and “weighted” to “weighed”

Corrected

Line 156: get rid of “the” before “both”

Removed

Line 160: change “more intense that” to “more intense than”

Corrected

Line 161: add “in” after “resulted”

Corrected

Line 166: missing comma after “sCD-IN”

Corrected: The legend has been rewritten.

Line 175: “high frequency” should have a hyphen

Corrected

Lines 184-185: “such as, for example” is redundant

Sentence modified

Line 198: missing comma after “N1s regions”

Corrected

Line 200: missing comma after “399.2”

Corrected

Line 202: missing coma after “286.2”

Corrected

Line 206: change “that could” to “which could”

Corrected

Line 209: add “in” after “resulted”

Corrected

Line 216: change “that determined” to “which determined”

Corrected

Line 230: change “point out to some” to “point out some”

Corrected

Line 244: change “resulted different to” to “resulted in differences to”

Corrected

Line 248: change “160 nm of diameter” to “160 nm diameter”

Corrected

Line 305: change “prevented of spraying” to either “protected from spraying” or “prevented by spraying” (not sure what the sentence is trying to say

Corrected to “protected from spraying”

Line 308: add article in front of “consequence”

Added “a”

Line 320: is “microcopy” supposed to be “microscopy”?

Yes, corrected to microscopy

Lines 333/334: is “microcopy” supposed to be “microscopy”?

Corrected

Line 337: missing article in front of “higher resolution”

Added “a”

Line 354: change “made us to prefer” to “led us to prefer” or something similar

Corrected

Line 356: add hyphen in “Teflon lined”

Corrected

Line 372-374: sentence here doesn’t make sense, I think it’s missing a verb

We have rewritten this sentence to make it more understandable. It was certainly quite awkward before.

Line 377: change “resulted similar to” to “had similar results to”

Changed

Lines 382-384: reword sentence, potential rewording- “Therefore, our products showed fluorescence when illuminated with UV light, which was indicative of the presence of carbon dots, as none of the reaction precursors fluoresce when illuminated with UV light.”

Sentence rewritten

Line 391: missing comma after “O”

Corrected

Line 393: correct subject-verb agreement by changing “indicate” to “indicates”

Corrected

Line 399: correct subject-verb agreement by changing “point out” to “points out”

Corrected

Line 401: “The quantum yield of the CDs obtained in this work resulted low” doesn’t make sense

Sentence rewritten

Line 402: add “that” after “reported”

Sentence rephrased

Line 407: correct subject-verb agreement by changing “has” to “have”

Corrected

Line 408: change “they resulted suitable” to “they had suitable results”

Rephrased

Line 420: add hyphen to “chitosan derived”

Added

Line 420: correct subject-verb agreement by changing “has” to “have”

Corrected

Line 428: change “resulted” to “were”

Corrected

Line 440: missing comma after “better solubility”

Corrected

Lines 443-444: The last half of the sentence seems to not be complete, maybe missing a verb?

Rephrased sentence

Line 445: “prevented to reaching the spraying” doesn’t make sense

Sentence corrected

Line 461: I think “higher amount” should be “higher amounts”

Yes, corrected.

Line 465: correct subject-verb agreement by changing “moves” to “move”

Corrected

Line 471: change “resulted lost” to “resulted in loss”

Corrected

Line 485: change “external” to “externally”

Corrected

Line 486: change “is being object” to “is an object”

Corrected

Line 488: correct subject-verb agreement by changing “has” to “have”

Corrected

Line 491: I think “cucumber plant” should be plural

Corrected

Line 498: missing comma after “fungi”

Corrected

Line 508: missing comma after “2,000”

Corrected

Line 511: change “were flowed through” to “were filtered through”

Corrected

Line 515: missing comma after “lyophilized”

Corrected

Line 519: change “weighted” to “weighed” and add comma after

Corrected

Line 552: add an article in front of “template”

Corrected

Line 563: missing comma after “charge”

Corrected

Line 564: missing comma after “CDs”

Corrected

Line 577: confused by the phrase “spraying was made”

Changed “made” for “applied”

Line 585: missing comma after “obtention”

Corrected

Line 600: change “derived of” to “derived from”

Corrected

Line 601: missing comma after “in this section”

Corrected

Line 618: missing comma after “fluorochrome”

Corrected

Line 620: missing an article before “manufacturer’s”

Corrected

Line 626: change “resulted similar to” to “had similar results to”

Rephrased

Line 635: “with respect to the derived from naked dsRNA” doesn’t make sense

Rephrased sentence

Reviewer 2 Report

The manuscript describes the synthesis of nanoparticles ("carbon dots") from glucose or saccharose and branched polyethylenimines at elevated temperatures. These particles have an averaged hydrodynamic diameter of 4 and 5 nm, respectively, and are positively charged according to extensive biophysical analysis. They electrostatically bind double-stranded RNA of about 464 bp (see below); the averaged hydrodynamic diameter of the composites increased to 350 and 160 nm for glucose and saccharose carbon dots, respectively. The RNA in RNA-coated nanoparticles was not protected against degradation by RNase A. Next, either the dsRNA or the RNA-coated glucose-based particles were sprayed by an airbrush to small cucumber plants to induce small interfering RNAs (siRNA). The amount of long RNA in a distal leaf three days after treatment was about  thousand times higher for RNA-coated particles than for naked RNA; siRNAs were 12fold more abundant with RNA-coated particles than with naked RNA. Thus, the described carbon dots plus airbrush spraying are a competent vehicle to induce RNAi in plants.

I would generally prefer the use of non-abbreviated word prior to an abbreviation. Examples are "RNAi (RNA interference)" (line 29), "dsRNA (double-stranded RNA)" (line 32), and "MRI (magnetic resonance imaging)" (line 103). 

Figure 4: For part A, the legend states a scale of 200 nm of the picture, while the inset in the figure states a scale of 20 nm. Which is correct?
Would it be possible to mark in A the section shown in B? That is, to me it is not obvious if B shows an enlargement of the noisy background of A or one of the "blobs" in A. Anyway, in these figures I do not see any particles of 5 nm diameter.

Lines 278, 317, 508, 514, 528, 560: "double distilled water", "double-distilled water", "distilled water", "MilliQ water": Are all these the same? And you really used double distillation and not some type of ion exchange plus further treatments?

Lines 227ff, Figures 5 and S4: 
Why does the pure dsRNA show a very large size distribution (250--1000 bp according to Fig. S4; 10--7000 nm hydrodynamic radius according to Fig. 5D)? This contradicts a homogeneous dsRNA of defined length as expected by the synthesis method. Why is this large size distribution not seen in Figure 6?

The English should be improved. The manuscript contains a lot of strange sentences.

Typos and further questions
=======================
Line 10: "carbon dots" => "carbon dots (CD)"
Line 12: "TEM" => "transmission electron microscopy"
Line 30: "progress have" => "progress has"
Line 33: "throughout" => "through"
Line 33: What are "DICER" and "RISC"?
Line 36: "RNA interference (RNAi)" => "RNAi" (already defined in line 29)
Line 45: "dsRNAs]," => "dsRNAs),"
Line 46: "[14,15, 16]" => "[14--16]"
Line 74: "N. benthamiana" in italics
Line 90, 110, 120, 625: "solvo hydrothermal" => "solvo-hydrothermal"
Line 94: "polyethyleneimines" => "polyethylenimines" (correct throughout the manuscript)
Line 96: "incorporation of N to the CD backbone and NH2 to the particle surface generating strong fluorescence" What are "N" and "NH2"? How does the "N" atom or the imino group generate strong fluorescence?
Line 106: Is this "CD-coated dsRNA" or RNA-coated CD?
Line 111: What is "bPEI"? (defined in line 123)
Line 160: What is "QY"?
Line 160: What is "XPS"?
Line 167: "glucose (A) and saccharose (B)" => "glucose (left) and saccharose (right)"
Figure 1B: Is one curve missing in both subfigures?
Line 219: "TEM" => "transmission electron microscopy (TEM)"
Line 258: "dsRNA results degraded" => "dsRNA degraded"
Line 273: "nanocomposite coated with the gCDs" => "gCD nanocomposites coated with dsRNA"
Figure 7: Which points are marked by crosses?
Figure 8: What is the experimental difference between pictures in the two top rows and the two bottom rows? 
Line 462_ "Alt-hough" wrong hyphenation
Line 526: "transmission electron microscopy (TEM)" => "TEM"
Line 561: "that it is immediate" => "that is immediate"
Line 625: What is meant by "solvo"? 
Line 642, Fig. S2: "weight: weight" => "weight:weight"
Figure S1: What is the dilution factor? 
Figure S3: Labels for x-axes are missing.

Author Response

(We acknowledge very much this revision and the indications and suggestions that have indeed helped us to improve our manuscript: we include our comments and answers in blue)

The manuscript describes the synthesis of nanoparticles ("carbon dots") from glucose or saccharose and branched polyethylenimines at elevated temperatures. These particles have an averaged hydrodynamic diameter of 4 and 5 nm, respectively, and are positively charged according to extensive biophysical analysis. They electrostatically bind double-stranded RNA of about 464 bp (see below); the averaged hydrodynamic diameter of the composites increased to 350 and 160 nm for glucose and saccharose carbon dots, respectively. The RNA in RNA-coated nanoparticles was not protected against degradation by RNase A. Next, either the dsRNA or the RNA-coated glucose-based particles were sprayed by an airbrush to small cucumber plants to induce small interfering RNAs (siRNA). The amount of long RNA in a distal leaf three days after treatment was about  thousand times higher for RNA-coated particles than for naked RNA; siRNAs were 12fold more abundant with RNA-coated particles than with naked RNA. Thus, the described carbon dots plus airbrush spraying are a competent vehicle to induce RNAi in plants.

I would generally prefer the use of non-abbreviated word prior to an abbreviation. Examples are "RNAi (RNA interference)" (line 29), "dsRNA (double-stranded RNA)" (line 32), and "MRI (magnetic resonance imaging)" (line 103).

Corrected in all cases

Figure 4: For part A, the legend states a scale of 200 nm of the picture, while the inset in the figure states a scale of 20 nm. Which is correct?

Would it be possible to mark in A the section shown in B? That is, to me it is not obvious if B shows an enlargement of the noisy background of A or one of the "blobs" in A. Anyway, in these figures I do not see any particles of 5 nm diameter.

The correct scale in A was in the picture, not in the legend. Yes, we chose this picture because the particles were well defined with respect the background, but did not take care in selecting them for this image. We now include a different picture in A of the carbon dots made in the same microscopy session. In B, the image is from other part of the same preparation where the regular lattice was clearly seen and differentiated with respect to the background.

Lines 278, 317, 508, 514, 528, 560: "double distilled water", "double-distilled water", "distilled water", "MilliQ water": Are all these the same? And you really used double distillation and not some type of ion exchange plus further treatments?

Yes, we used always MilliQ water, including the 0.22 um filtering and the UV decontamination in all the steps involving CD synthesis, purification, dilution, elution and the combination with the dsRNAs. We incorrectly described double-distilled water when in fact we mean MilliQ water. So, we have corrected all the sentences regarding the type of water used accordingly.

Regarding the washing step of the plant leaves were washed after the dsRNA applications, we did with water purified though a ion exchange column, that we guess is equivalent to distilled water.

Lines 227ff, Figures 5 and S4:

Why does the pure dsRNA show a very large size distribution (250--1000 bp according to Fig. S4; 10--7000 nm hydrodynamic radius according to Fig. 5D)? This contradicts a homogeneous dsRNA of defined length as expected by the synthesis method. Why is this large size distribution not seen in Figure 6?

In Fig. S4 the dsRNAs in the agarose gels both in naked form and coated with the gCDs were FITC-labelled. For unknown some reason, this labelled dsRNA showed a range of sizes in the agarose electrophoresis instead of a single clear band. We do not have a good explanation for this. Nevertheless, the labelled dsRNA was not used in the hydrodynamic diameter analyses.

Actually, in Fig. 6A-C the naked dsRNA and the CD-coated dsRNA, showed well defined bands. In the hydrodynamic diameter measurement of the dsRNAs (Fig. 5C) a main peak appears at a maximum of 40 nm, that should be the most frequent state of dsRNA in solution, and other density peaks appeared at >1000 nm, that should correspond to a fraction of aggregated dsRNA molecules in the water solution. In the electrophoresis, the electric current separates the molecule aggregates and single bands appear. In any case, the dsRNA aggregates are a minoritarian fraction of the state of the dsRNA in solution.

The English should be improved. The manuscript contains a lot of strange sentences.

We have carefully revised the English and rephrased many sentences.

Typos and further questions

=======================

Line 10: "carbon dots" => "carbon dots (CD)"

Corrected

Line 12: "TEM" => "transmission electron microscopy"

Corrected

Line 30: "progress have" => "progress has"

Corrected

Line 33: "throughout" => "through"

Corrected

Line 33: What are "DICER" and "RISC"?

Corrected.

Line 36: "RNA interference (RNAi)" => "RNAi" (already defined in line 29)

Corrected.

Line 45: "dsRNAs]," => "dsRNAs),"

Corrected.

Line 46: "[14,15, 16]" => "[14--16]"

Corrected.

Line 74: "N. benthamiana" in italics

Corrected.

Line 90, 110, 120, 625: "solvo hydrothermal" => "solvo-hydrothermal"

Corrected.

Line 94: "polyethyleneimines" => "polyethylenimines" (correct throughout the manuscript)

Corrected

Line 96: "incorporation of N to the CD backbone and NH2 to the particle surface generating strong fluorescence" What are "N" and "NH2"? How does the "N" atom or the imino group generate strong fluorescence?

Yes, this sentence needed correction. We have removed the term “fluorescence” in this paragraph, because the fluorescence is an intrinsic characteristic of carbon dots with crystalline structure and has nothing to do with the passivation with nitrogen compounds. On the other hand, we have corrected NH2 for -NH2.

Line 106: Is this "CD-coated dsRNA" or RNA-coated CD?

This is an interesting question: we think that it its more proper to talk about dsRNA-coated, as we imagine that the dsRNA molecules are coated with more or less CDs, depending on the concentration of CDs in the solution. We think of the dsRNA-CD nanocomposite like a long-beaded necklace.

Line 111: What is "bPEI"? (defined in line 123)

Corrected

Line 160: What is "QY"?

Corrected

Line 160: What is "XPS"?

Corrected

Line 167: "glucose (A) and saccharose (B)" => "glucose (left) and saccharose (right)"

The legend of the figure has been corrected.

Figure 1B: Is one curve missing in both subfigures?

Well, we count 11 curves corresponding to the 11 wavelengths shown in the legends, used for the excitation of the CDs.

Line 219: "TEM" => "transmission electron microscopy (TEM)"

Corrected

Line 258: "dsRNA results degraded" => "dsRNA degraded"

Corrected

Line 273: "nanocomposite coated with the gCDs" => "gCD nanocomposites coated with dsRNA"

Corrected

Figure 7: Which points are marked by crosses?

The arithmetic mean. Now indicated in the figure legend.

Figure 8: What is the experimental difference between pictures in the two top rows and the two bottom rows?

The two top pictures were taken from leaves immediately after the spraying of dsRNA or gCD-dsRNA. In the two bottom rows the pictures were taken after the washing of the leaves. It could be seen that the leaves where we applied gCD-dsRNA, much more fluorescence remained after the washing than in the leaves were only naked dsRNA was applied. For clarification we put in the figure, on the left, an indication of the situation (before washing/after washing).

Line 462_ "Alt-hough" wrong hyphenation

Corrected

Line 526: "transmission electron microscopy (TEM)" => "TEM"

Corrected

Line 561: "that it is immediate" => "that is immediate"

Corrected

Line 625: What is meant by "solvo"?

Corrected to solvothermal.

Line 642, Fig. S2: "weight: weight" => "weight:weight"

Corrected

Figure S1: What is the dilution factor?

1:5, now included in the legend.

Figure S3: Labels for x-axes are missing.

Corrected to include the pH label in the x-axes.

Round 2

Reviewer 1 Report

All issues have been addressed